# Gallato Zirconium (IV) Phtalocyanine Complex Conjugated with SiO_2_ Nanocarrier as a Photoactive Drug for Photodynamic Therapy of Atheromatic Plaque

**DOI:** 10.3390/molecules26020260

**Published:** 2021-01-06

**Authors:** Yuriy Gerasymchuk, Wojciech Kałas, Jacek Arkowski, Łukasz Marciniak, Dariusz Hreniak, Edyta Wysokińska, Leon Strządała, Marta Obremska, Larysa Tomachynski, Viktor Chernii, Wiesław Stręk

**Affiliations:** 1Institute of Low Temperature and Structure Research, Polish Academy of Sciences, ul. Okólna 2, 50-422 Wrocław, Poland; l.marciniak@intibs.pl (Ł.M.); d.hreniak@intibs.pl (D.H.); w.strek@intibs.pl (W.S.); 2Ludwik Hirszfeld Institute of Immunology and Experimental Therapy, Polish Academy of Sciences, Rudolfa Weigla 12, 53-114 Wroclaw, Poland; wkalas@konto.pl (W.K.); edyta.wysokinska@hirszfeld.pl (E.W.); leon.strzadala@hirszfeld.pl (L.S.); 3Department of Preclinical Studies, Wrocław Medical University, ul. K. Bartla 5, 51-618 Wrocław, Poland; jacekarkowski@poczta.onet.pl (J.A.); marta.obremska@umed.wroc.pl (M.O.); 4V.I. Vernadskii Institute of General and Inorganic Chemistry, 32/34 Palladin Ave., 03-142 Kyiv, Ukraine; l.tomachynska@gmail.com (L.T.); v.chernii@gmail.com (V.C.)

**Keywords:** axially substituted, water soluble, zirconium phthalocyanines, SiO_2_ nanocarriers, photosensitizers, ROS generation, photodynamic therapy, atherosclerosis, cell cultures, macrophages

## Abstract

A new conjugate of gallato zirconium (IV) phthalocyanine complexes (PcZrGallate) has been obtained from alkilamino-modified SiO_2_ nanocarriers (SiO_2_-(CH_2_)_3_-NH_2_NPs), which may potentially be used in photodynamic therapy of atherosclerosis. Its structure and morphology have been investigated. The photochemical properties of the composite material has been characterized. in saline environments when exposed to different light sources Reactive oxygen species (ROS) generation in DMSO suspension under near IR irradiation was evaluated. The PcZrGallate-SiO_2_ conjugate has been found to induce a cytotoxic effect on macrophages after IR irradiation, which did not correspond to ROS production. It was found that SiO_2_ as a carrier helps the photosensitizer to enter into the macrophages, a type of cells that play a key role in the development of atheroma. These properties of the novel conjugate may make it useful in the photodynamic therapy of coronary artery disease.

## 1. Introduction

Coronary artery disease remains one of the major causes of morbidity and mortality in developed countries. In spite of enormous progress in diagnostic and therapeutic modalities there are still several unresolved problems. One of them is the identification and stabilization of vulnerable (prone to rupture) atherosclerotic plaques. Moreover, even if the culprit coronary lesion is successfully identified, stented and blood flow is restored, re-narrowing of the artery (restenosis) may occur after several weeks. Vascular smooth muscle cells (SMC) play a key role in this phenomenon. 

Targeting macrophages, key cells in development of atherosclerotic plaque, seems to be a very promising area in both the diagnosis and prevention of atherosclerosis, especially in the form of vulnerable plaques [1,2]. In both areas photodynamic therapy (PDT) may be an interesting option. Various ligand–receptor systems are currently being evaluated in terms of targeting functionalized nanoparticles. Antibody-based systems are the most often studied, but other solutions such as antibody fragments, peptides or small molecules have also been tested [3,4].

It has also been shown that PDT effectively induces cell apoptosis in human vascular smooth muscle cells. This effect, when applied in clinical practice, may provide an alternative solution for the treatment of in-stent restenosis [5]. Another example of light-sensitive materials used in the treatment of atherosclerosis are photoactivated titania-based coatings proposed as for cardiovascular stents [6]. The primary limitation in the application of the PDT method is the depth of tissue penetration by laser radiation [7]. It is a well-known fact that tissues scatter and absorb less red and infrared light in three so-called biological windows ranging from around 650 to 1850 nm [8,9,10]. Therefore, there is an ongoing search for photosensitizers active in that wavelength range, which are both quickly taken up by the target tissue and eliminated from the organism [7]. 

One of the many photosensitizers that meet these criteria are phthalocyanins (Pc) and their derivatives. Macrocyclic complexes, especially porphyrins and phthalocyanins, are widely used as dyes in different fields of bio-applications [11]. The phthalocyanine dyes are activated by near infrared light (in the biological window), have high coefficients of emission and are photoreducible (as catalysts in photooxidation reactions). 

Compounds from this group have good photophysical properties and have absorption maxima in near infrared (over 700 nm) [9]. However, due to their high hydrophobicity they are difficult to introduce into biological systems. For example, zinc phthalocyanine complex, for which a high level of biological activity has been demonstrated, requires encapsulation in liposomes to effectively enter tissues [10].

The main limitations of the use of phthalocyanine pigment in biological systems, especially in photodynamic diagnostics and therapy, are the hydrophobicity of phthalocyanine macrocycle and consequent aggregation of phthalocyanine compounds in an aqueous medium (biological liquids). The lateral substitution of phthalocyanine rings with hydrophilic groups (such as sulphonic or carboxylic groups) is the technique most commonly-used to solve the first problem. In several countries (e.g., France, Russia, Canada) a formulation based on tetra-sulfonated aluminum phthalocyanine (“Photosens”) was used in clinical practice [9]. Unfortunately, this type of substitution impairs the spatial structure of phthalocyanine complexes and deteriorates the optical properties of these complexes. Another solution—axial substitution of the central metal atom of phthalocyanine complexes causes the central metal atom to be located outside the macrocyclic plane. It can coordinate (for metals with a coordination number higher than 6, e.g., Zr has coordination number 7 or 8) additional ligands—solvent molecules. On the other hand, this type of coordination results in a high tendency toward the agglomeration of phthalocyanine complexes in a hydrophilic medium [12,13].

Apart from appropriate photophysical properties of the photosensitizer, another important issue is how the compound is delivered to biological systems (e.g., cells, tissues). A suitable carrier should be used, preferably one that is selective enough to deliver the photosensitizer to (and only to) the target tissue. Researchers from Case Western Reserve University, as the first scientific group in the world, presented the mechanism of delivering photosensitizers to tissues by immobilizing them on nanoparticles (metallic Au) used as a carrier [14,15]. Preliminary studies have shown that anti-cancer drugs, associated with the surface of the gold nanoparticles, begin to accumulate in the tumor within minutes of their introduction and can be activated for an effective treatment period of two hours. An accumulation of the same preparation that is introduced without the carrier lasts two days, and the preparation mainly attacks the surface of the tumor. As an example of a different carrier type, a liposomal formulation of PVP-conjugated chlorin e6 being used for photodynamic reduction of atherosclerotic plaque has also been reported [16].

Promising results were also obtained with silica-based matrices. Small silica particles in an isolated rat heart model were successfully delivered to coronary circulation and were shown to have no harmful effects on the heart muscle. [17]. Another study showed that the porous silica matrix successfully delivers photosensitizer to smooth muscle cells and enables PDT (phototoxic reaction) [18]. The use of alkyloamino-modified SiO_2_ nanocarriers for the transportation of biologically active substances in blood vessels and into the cells is also a known method [17,18].

Composites of SiO_2_ nanocarriers with Pc complexes have been described earlier [19,20] but only for their catalytic activity and with laterally conjugations of the phthalocyanine macrocycle. In this work, we propose a novel way to conjugate axially substituted zirconium-phthalocyanine complex to alkyamino-modified carrier, through a carboxylic group of the axial ligand. In contrast to previously used approaches, we used the cheaper and better catalyst DCC (N, N’-dicyclohexylcarbodiimide) instead of N-(3-dimethylaminopropyl)-N’-ethylcarbodiimide hydrochloride, which was proposed by other authors [20]. We assumed that the settlement of the complex on the surface of SiO_2_ nanocarrier would allow us to avoid agglomeration of the complexes observed during their introduction into hydrophilic environment, as well as the homogenous doping of silica, when agglomeration processes are also taking place [14]. In this case there is a disturbance of the electronic structure similar to that which occurs with the lateral substitution of the phthalocyanine ring. Moreover, embedding the pigment on a carrier facilitates its transport in the organism, allowing it to reach the target cells and reduce the toxicity of phthalocyanine. 

The main aim of this work was to synthesize and investigate hybrid materials composed of water soluble, axially gallato substituted zirconium phthalocyanine complex and nanosized SiO_2_ carriers for biomedical applications [14,20,21]. Biostatic properties of axially substituted zirconium-phthalocyanine complexes have been shown in our earlier work [21]. The carriers should be able to pass through blood vessels, therefore SiO_2_ particles with an average grain size of less than 50 nm were chosen. The surface of the particles should be modified with amino groups for future attachment of phthalocyanine complexes [22]. In the next step, selected macrocyclic complexes were attached to the carrier by covalent bonding, which was split by chemical agents or enzymes (in vivo). For this purpose, an amide (peptide) bond existing between the carboxyl of axial ligand in the phthalocyanine complex and amino groups on the surface of the carrier was used. The photophysical properties of a free phthalocyanine complex in saline solution and two hybrid materials based on this complex, were compared. Due to the binding of the dye on the surface of the carrier, the process of aggregation of the dye molecules was significantly limited, resulting in enhanced luminescence. In particular, the effect of binding macrocyclic complexes to the surface of carriers on the absorption and emission spectra of the materials obtained was investigated. The obtained results proved that the studied materials are efficient photocatalysts (exhibiting controlled singlet oxygen generation) and, when used with silica nanocarriers, can be a promising method for photodynamic diagnostics and atherosclerosis therapy.

## 2. Results and Discussion

### 2.1. The Structure and Morphology of Obtained Conjugates

The structure of the complex compound PcZr (Gallate) was confirmed using ^1^H NMR spectroscopy (Pc ring—9.38(m, 8H); 8.18(m, 8H)—corresponded to Ha and Hb protons of Pc-cycle, respectively, and signals of protons in axial ligand (gallate fragment) —7.38 − 7.21(d, 1H, Ph) 7.04 − 6.97(d, 1H, Ph)) and ESI mass spectroscopy (molecular ion signal at *m*/*z* = 771,8978. THe the signal with *m/z* = 916.0863, which corresponds to the PcZr(Gallate)@2DMF molecule, was two times more intensive, which is consistent with the theory presented in our earlier works that the zirconium atom, having a characteristic coordination number of 8, in such a system, where it is drawn out of the macrocycle plane by the axial ligand, is able to coordinate additional solvent molecules, which ensures the solubility of this type of phthalocyanine complex in water and polar organic solvents). The obtained results correlate well with those we obtained earlier for this complex [13,14], and confirm the correctness of the synthesis. The ICP-OES tests showed that the zirconium content in the conjugate was 1488 mg/kg (ppm), and they also showed the presence of a hafnium admixture at the level of 2.5 mg/kg (ppm). After converting these data to the molar mass of the phthalocyanine complex, it gives us a complex content in the conjugate of about 12.49%.

According to the TEM and SEM images (Figure 1a,b) the grain size of mono-dispersed PcZrGallate-SiO_2_ Nps, obtained by both methods A and B, were about 40–50 nm and had a very narrow grain size distribution and a low degree of agglomeration of nanoparticles. However, for spectroscopic studies, the suspensions were all but dispersed with a high power dispergator to obtain a stable suspension. 

Due to the low content of pigment in the material, the characteristic bands of PcZr-gallate complex were shielded by very broad and intense bands derived from the silica-modified amino groups (Figure 2). In the MIR spectra, for PcZr(IV) gallate complex, absorption bands corresponding to valence vibrations of C = O group in 1720–1700 cm^−1^ region, υ (C–O)—1220–1200 cm^−1^, υ (O–H)—3600–3550 cm^−1^ were observed. IR spectra approving of the pigment substitution of surface aminogroups, which are manifested where the band decreases, related to vibration of free –NH_2_ groups, and the erasure of the wide single band with a maximum width of about 1656 cm^−1^, which, according to [21], is related to the vibration of ν(C = O) group and amidic band ν(CO)+ν(CN) [20,21].

### 2.2. Spectroscopic Studies of Free Phthalocyanine and Its Conjugate with SiO_2_ Nanocarrier in Saline

In the absorption spectra one can observe the bands corresponding to the phthalocyanine complex: the Soret band (B-band) with a maximum wavelength at 342 nm, and the Q-band with the maximum wavelength 685 nm. In the Q-band region, the low intensity satellite band with a maximum at 619 nm was observed.

For the samples obtained by the B-method, the concentration of the complex is lower because of a lack of amino groups contained in the closed pores of the material, and it is proved by the lower intensity of absorption (Figure 3). On the other hand, all pigment molecules were bonded with the surface of SiO_2_ nanoparticles. This fact was proved by the observation that the Soret band was very intensive, but it fully repeated the shape of the band of undoped silica in this region, and by the shift of Q-band maximum toward longer wavelengths and the lack of a clear band–band satellite in the range of Q. This phenomenon has been described by us earlier [13].

Part of the pigment included in the material by the a-method is not bound to the surface but closed in the pores of silica, and therefore differences arise in the shape and intensity of the absorption spectra for the materials obtained in various ways [13].

The zirconium phthalocyanine complex with gallic acid as an axial substitute was not chosen accidentally. Firstly, it was chosen because of its solubility in water. Besides, an interesting effect of this complex was previously registered: it consists of a rapid increase in the intensity of light absorption upon high-power irradiation (xenon lamp 150 W), wherein the intensity increased when increasing the exposure time of the sample, and after reaching the plateau, the effect persisted for a very long time (up to several weeks). Similar changes have been described by our group and other authors, and were attributed to a photoreduction reaction of the phthalocyanine π system as a result of the impact of emerging singlet oxygen molecules, or free radicals, because such a mechanism is also considered to be a possible type of impact on cell photosensitizers in photodynamic therapy [23,24,25]. For further use of these systems for biological research, it seemed logical to look for this effect in our materials at an excitation light wavelength of 620 nm, corresponding to the maximum of the excitation band for the phthalocyanine (Figure 4). For this purpose, we used a 4 W 620 nm diode. Unfortunately, there was no significant increase in absorption even at fairly large exposures. On the other hand, the effect of the increase in absorbance, even at short exposure times, was registered using ultraviolet and infrared lamps in a wide range. The spectra of all used irradiation sources are shown in the Figure 2, and the spectra of our composites before and after irradiation with different light sources are shown below. Following the analysis of the absorption spectra, one can draw the following conclusions [13,14]: in the absorption spectra of composite materials prepared by both the methods, the characteristic bands of phthalocyanine in an aqueous medium can be observed. As was shown in our earlier works [13,14], the shape of the spectra (a wide Q band with a low intensity of absorption with the maximum shifted to high energy wavelengths) is characteristic of phthalocyanines in aqueous media or for complexes incorporated in silica gel blocs (especially in the B-band region) and is related to the partial dimerization of the complexes. However, the emission was not observed in the agglomerated phthalocyanine system. The fact that immobilization of the phthalocyanine complex at the surface of SiO_2_ nanocarrier leads to the efficient and strong emission of phthalocyanine (Figure 4) means that the phthalocyanine molecules are not aggregated.

The analysis of the emission spectra of materials shows the next interesting effects associated with the presence of the carrier in the system. While the free phthalocyanine, which is not bonded to the support in aqueous media, has no emission bands that are observed for the non-aqueous solutions such as DMSO and methanol, the phthalocyanine complexes supported by a carrier have demonstrated a quite intense emission in aqueous suspensions (suspension in saline). In our opinion this is, in the case of free phthalocyanine, the effect of “face-to-face” dimerization of phthalocyanine rings and further agglomeration into larger units in aqueous media, which leads to the coupling of systems and the mutual quenching of emissions. Dye linked with the SiO_2_ carrier has the limited opportunities for dimerization and is characterized by a fairly intense emission band with a maximum of around 705 nm at the excitation wavelength of 620 nm. It means that this affects the efficiency of the application of this phthalocyanine dye for use in photodynamic therapy approaches. 

#### Reactive Oxygen Species (ROS) Generation in DMSO

Under NIR light irradiation, the PcZr(IV)gallate@SiO_2_ composite (obtained by B- method) in the form of a suspension in DMSO is characterized by a very intense ROS generation as shown by a result of the DPBF absorption measurements shown in Figure 5. A summary of the graphs of the dependence of the absorbance intensity in the DPBF band maximum (λ_max_ = 417 nm) on the red light irradiation time is presented in Figure 6. The graph for the same relationship, recorded for the zinc phthalocyanine (PcZn) solution, is provided as a reference. 

At the same time, we also observe an increase in the absorption intensity for the composite (the same as was observed in the suspension of the test composite in saline), which depends on the red light exposure time (Figure 7). Likewise, in a solution of the phthalocyanine complex in DMSO as well as in saline before that, such an increase was not noted (Figure 7).

### 2.3. Biocompatibility and Phototoxicity of PcZrGallate Preparations

For the biocompatibility study, the macrophage RAW264.7 cell line was selected due to its enhanced capability to phagocyte particulate matter (Figure 8). 

The nanoformulation of PcZrGallate was not toxic even in high doses 125 µg/cm^2^. Interestingly, the soluble form of PcZrGallate revealed dose-dependent dark cytotoxicity in concentrations higher than 100 μg/ml. For these reasons, the low concentrations were selected for photosensitization and phototoxicity studies. The two light sources were selected for inducing a phototoxic reaction in the 620 nm diode and the IR lamp. The soluble form of PcZrGallate was not able to sensitize the cells. The illumination of the cells photosensitized with soluble PcZrGallate did not lead to the generation of ROS or a decrease in the viability of the cells (Figure 9). On the other hand, if the cells were photo-sensitized with the nanoparticle formulation of PcZrGallate-SiO_2_, generation of reactive oxygen species [26] can be observed upon illumination with both sources of light. It should be noted that the quantity of ROS was sufficient to induce further loss of viability only when the cells were irradiated with the IR lamp. In the case of irradiation with the 620 nm diode laser, the induction of ROS was not accompanied by loss of viability. This suggests that the generation of insufficient amounts of reactive oxygen species or its different nature do not lead to the induction of cell death. 

The effective comparison of these two formulations is difficult and complex. On the one hand we have soluble (but with low solubility in an aqueous solution) PcZrGallate, where the dose is expressed in mass/volume concentration units, while on the other hand we have insoluble nanoparticles (but despite this insolubility, the agglomeration of Pc complex molecules in aqueous media was much lower due to their immobilization on the carrier). A sediment of particulate nanomaterials tends to form on the bottom of cell culture well close to the monolayer of the cells [27]. For this reason, the dosimetry of nanoparticles is complex. As most of the particles were located on the bottom of the cells, the dose was expressed as μg/cm^2^. Additionally, we need to appreciate that, contrary to the solution of PcZrGallate, silica nano-formulation consists of only a portion of the photo-active compound. Moreover, the mechanism of internalizations would be different, as the uptake of silica nanoparticles depends on endocytosis [28]. Despite such reservations, we have shown that silica PcZrGallate conjugate poses less unspecific toxicity even in high doses. Moreover, a relatively low dose of 300 ng/cm^2^ was sufficient to photosensitize the cells. The data presented here demonstrate that nano-formulation of PcGrGallate can be successfully used for the photosensitization of cells. It could be postulated that the uptake of nanophotosensitzer depends on the endocytic capability [29] of target cells and may be an example of a Trojan horse cytotoxic effect [30,31]. The nanoformulation is an example of a positive solution to the problem of enhancing the penetration of difficult to handle drugs and substances into the cells.

## 3. Materials and Methods

### 3.1. Synthetic Procedures

All used reagents with ≥99% purity were purchased from Alfa-Aesar (USA), Sigma-Aldrich (USA) and ABCR (Germany) and used without further purification.

The synthesis of PcZr (Gallate) (Figure 10a,b) was described in detail elsewhere [13,14]. Briefly, in the first step the dichlorozirconato phthalocyanine complex was synthesized from 1,2-dicyanobenzene and ZrCl_4_ in 2-methylnaphtalene solution, followed by the reaction of the ligand exchange in PcZrCl_2_, where Cl ligands were replaced by gallic acid in 1,2,4-trichlorobenzene with heating under reflux (5 h in 200 °C).

The inverted micelle method of preparation of non-agglomerated nanoparticles [16,20,21] was elaborated in order to obtain suitable SiO_2_ nanocarriers. A general procedure for their preparation was as follows: 12.3 ml of hexane was placed in an Erlenmeyer flask on a magnetic stirrer, and 1 ml of Triton X100, 1.04 ml of 1-hexanol and 0.6 ml of distilled and demineralized water were added. After 15 min of intensive stirring, 0.5 ml of tetraethyl orthosilicate (TEOS) and 0.3 ml of 25% solution of NH_4_OH were added dropwise, and the reaction mixture was left on a magnetic stirrer for 5 h. After that, the flask was capped and placed in the refrigerator for 3 days. The particles were precipitated with chilled acetone, washed several times with ethanol and water, and separated by centrifugation. For the modification with the pigment we used two different approaches. In the first one we used the direct modification of (3-Aminopropyl) triethoxysilane (APTES), and this conjugate was included in the inverted micelle synthesis of SiO_2_ nanoparticles (Sample A). In the second approach, firstly the TEOS/APTES nanoparticles were obtained, and then they were modified with PcZrGallate (Sample B).

#### 3.1.1. PcZrGallate-APTES and PcZrGallate-SiO_2_ Conjugates (Sample A)

The first step of this method was the preparation of PcZrGallate-functionalized silanization precursor [20]. To produce the covalent binding between PcZr(IV)Gallate and SiO_2_, PcZrGallate was first reacted with APTES in the presence of N,N’-dicyclohexylcarbodiimide (DCC) to form the PcZrGallate silanization precursor. In a typical procedure, 1.0 × 10^−5^ mol PcZrGallate (dissolved in anhydrous dimethylformamide, DMF), 1.6 × 10^−4^ mol DCC and 40 μL APTES were mixed in a round bottom flask, and reacted for 24 h while being stirred in an ice bath (5 °C) to form the PcZrGallate—functionalized silanization precursor.

The PcZrGallate-SiO_2_ conjugates were prepared by the co-hydrolysis of PcZrGallate-functionalized silanization precursor with TEOS in a W/O microemulsion. An appropriate amount of PcZrGallate-functionalized silanization precursor was added to the W/O reverse microemulsion prepared by mixing 12.3 mL of cyclohexane, 1 mL of Triton X-100, 1.04 mL of n-hexanol and 0.30 mL of water. After stirring for 15 min, 0.5 mL net TEOS was added drop by drop to the microemulsion and the mixture was stirred vigorously for 15 min in order to enter the TEOS into the hydrolysis reaction. Then 0.15 mL of concentrated ammonia solution was added to initiate the polycondensation reaction. After the solution was stirred for 5 h at room temperature, the mixture was placed in a refrigerator and aged for 3 days, then the nanoparticles were isolated by acetone, followed by centrifuging and washing them with ethanol and water several times to remove any surfactant molecules and unreacted materials. The size of the nanoparticles was primarily controlled by the molar ratio of water to Triton X-100 and the molar ratio of water to TEOS. The nanoparticles were dried in a vacuum for over 24 h at room temperature.

#### 3.1.2. PcZrGallate Modified SiO_2_ Nanoparticles (B-Sample)

In order to obtain alkylamino-modified SiO_2_ by the method described above, 1.0 × 10^−5^ mol PcZrGallate (dissolved in anhydrous dimethylformamide, DMF) and 1.6 × 10^−4^ mol DCC were added to a 50 ml round bottom flask. The flask containing the reaction mixture was put into an ice bath and left on a magnetic stirrer for 24 h. The modified NPs were separated by centrifuging and washing with ethanol and water several times to remove the unreacted dye.

### 3.2. Structure and Morphology Measurements

The 1H NMR spectra were registered by the 300 MHz AMX Bruker NMR spectrometer in DMSO-*d_6_* with TMS as per internal standards (δ 9.66 − 9.21 (m, 8H), 8.68 – 8.01 (m, 8H), 7.84 (s, 1H), 7.29 (d, *J* = 51.1 Hz, 1H), 7.01 (d, *J* = 20.1 Hz, 1H). PcZr (Gallate) was examined by ESI Mass-spectrometry using a Bruker Apex Ultra mass spectrometer with ESI ion source in N,N’-dimethylformamide (DMF) solution. The presence of PcZr(IV)gallate complex in the obtained material was approved by using IR spectroscopy (MIR) of material in KBr pellets (Spectrometer Bruker 113v FTIR). The content of phthalocyanine in the conjugate obtained by method A was determined by the ICP-OES on a 7000 iCAP ICP-OES by Thermo scientific. For ICP measurements, a probe of A-conjugate was dissolved in hydrofluoric acid in a microwave-assisted Teflon lined autoclave (Magnum II, Ertec, Poland).

The morphology of the obtained composite materials was investigated with Scanning Electron Microscopy (FESEM FEI Nova NanoSEM 230) and Transmission Electron Microscopy (FEI Tecnai G2 20 X-TWIN microscope).

### 3.3. Optical Spectroscopy Measurements

The UV-Vis-NIR absorption (in water (saline) and spectroscopic grade DMSO—molar concentration for free phthalocyanine complex was C_m_ = 1 × 10^−5^ M, concentration of the preparation C = 0,1 g/l) were measured by using the CARY 5000 UV-Vis-NIR spectrophotometer (Agilent). Te emission spectra were collected by using: Spectrophotometer FLS980 (Edinburgh Instruments), SD2000 CCD spectrophotometer (OceanOptics) and THR1000 monochromator (JobinYvon) and using 1 cm quartz cuvettes. All emission measurements were carried out for samples of Cm = 5 × 10^−6^ M and C = 0,1 g/l. The spectra of the excitation light sources used are presented on Figure 11. The suspension of conjugates for all optical measurements were prepared by ultrasonification in a UZDN M900-T (22 kHz, 900 W) Akadempribor, Ukraine dispergator. 

#### Reactive Oxygen Species (ROS) Generation Measurements

Absorption spectroscopy (Agilent CARY 5000 UV-Vis-NIR spectrophotometer) was also used for analysis of the singlet oxygen (ROS) generation of the free phthalocyanine solution and composite suspension prepared by sonification a 50 mg of conjugate in 10 mL of DMSO during 30 min with 900 W–22 kHz dispergator UZDN-M-900-T (Ukrrospribor, Ukraine); then 2 ml of supernatant was transferred with an automatic pipette to a quartz cuvette, and measurements were carried out in the presence of 1,3-diphenylisobenzofuran (DPBF) as an ROS marker due to the method described in [32,33]. Zinc phthalocyanine (ZnPc) was used as an internal standard. In the case of PcZn and PcZr (Gallate), the molar concentration of the tested solutions was in the order of 10–5 mol/L. In case of PcZr(Gallate)-SiO2 conjugate, the concentration was 0,5 mg/mL, which means that the concentration of the phthalocyanine complex, calculated on the total volume of the suspension, was approximately half that of the real solutions. Additionally, a slight sedimentation of the composite was observed during the measurements, therefore the graphs were normalized. The graphs of the absorbance intensity versus the exposure time of the samples have also been normalized for visualization and mutual comparison.

### 3.4. Cell Culture

The studies were carried out using mouse monocyte/macrophage cell line (RAW264.7). The cells were cultured in Dulbecco Modified Eagle’s Medium (DMEM; HyClone, Waltham, MA, USA) or DMEM without phenol (DMEM, IITD) supplemented with 10% Fetal Calf Serum (Gibco, Carlsbad, CA, USA), 3% glutamine, 10 mM HEPES, 1 mM sodium pyruvate and with Antibiotic and Antimycotic Solution (Sigma-Aldrich, St. Louis, MO, USA). The cells were maintained at 37 °C, 5% CO_2_, and 95% humidity (NuAire, Plymouth, MN, USA). 

### 3.5. Cell Treatment Protocols

#### 3.5.1. Detection of Reactive Oxygen Species

For detection of reactive oxygen species Reactive Oxygen Species (ROS), Dichlorodihydrofluorescein Diacetate Detection Reagent was used (Invitrogen, Paisley, UK). The oxidation of probes was detected by monitoring the increase in fluorescence (490 nm/510 nm). Before each experiment, RAW264.7 cells were seeded for 24 h on 96-well plates at 10 × 10^3^ cells per well in the medium of free of phenol red and other colorimetric dyes. The RAW264.7 cell line was incubated with photosensitizers for 4 h. Then the growing cell media were removed and the cells were incubated with dye resuspended in PBS (10 µM) for 30 min. Then the cells were washed in growth media and baseline fluorescence intensity was determined. The level of ROS production was measured after exposure to a 620 nm diode or IR lamp and incubated for 15 min on a VallacVictor2 plate reader (Perkin Eliner). Negative controls were performed according to the manufacturer’s protocol. The average ± SD from at least three independent experiments are shown on graphs.

#### 3.5.2. Cell Cytotoxic Assay

After PS treatment, cytotoxicity was determined using the CellTiter 96 AQueous Non-Radioactive Cell Proliferation Assay (Promega, Madison, WI, USA). RAW264.7 cells were seeded on 96-well plates for 24 h and then treated with the indicated concentration of PcZrGallate or PcZrGallate-SiO. If necessary, cells were irradiated and incubated for the next 24 h. To exclude the influence of reagents, the properties of 490-nm absorbance were tested after irradiation or photosensitizer treatment. Then, cells were incubated for 1.5 h with 10 µL MTS reagent at 37 °C. 490-nm absorbance corresponding to the number of metabolically active cells was measured on a VallacVictor2 plate reader. The loss of the viability was calculated as the percentage difference of the control sample (100%). The average ± SD from at least three independent experiments is shown on the graphs.

## 4. Conclusions

The preparation of a new conjugate composed of gallato zirconium (IV) phthalocyanine complex, which was bonded to propylamino modified SiO_2_ nanocarriers through metal coordinated axial ligand (by two different methods) was described for the first time. It was shown that a method that used the initial interaction of APTES with the carboxylic group of the action ligand, with the subsequent incorporation of this conjugate in the microemulsive synthesis of silica nanoparticles, was more effective. The structure, morphology and spectroscopic properties of obtained materials were investigated in detail. It was demonstrated that such conjugates are characterized by enhanced emission yield due to restricted agglomeration of phthalocyanine molecules and intensive ROS generation in DMSO and saline under the influence of IR irradiation. This positively verifies their predisposition to use in photodynamic therapy. This assumption was further confirmed by in vitro tests on cell cultures, which showed that PcZrGallate-SiO_2_ conjugates have a better profile of dark cytotoxicity over soluble PcZrGallate complexes. PcZrGallate-SiO_2_ conjugate induces a photocytotoxic reaction revealed by ROS production and can effectively sensitize RAW264.7 cells to IR light, and in addition, SiO_2_ nanoformulation enhances PcZrGallate’s ability to photosenitize cells.

The novel conjugate presented in the paper may be an interesting option for PDT, especially in the treatment of unstable atherosclerotic plaques and the prevention of restenosis. It is well tolerated by cells and has a cytotoxic effect when irradiated with IR light. Further research is justified to investigate its effects on macrophages and vascular smooth muscle cells—potential targets for photodynamic diagnostic and treatment of atherosclerosis. 

## Figures and Tables

**Figure 1 molecules-26-00260-f001:**
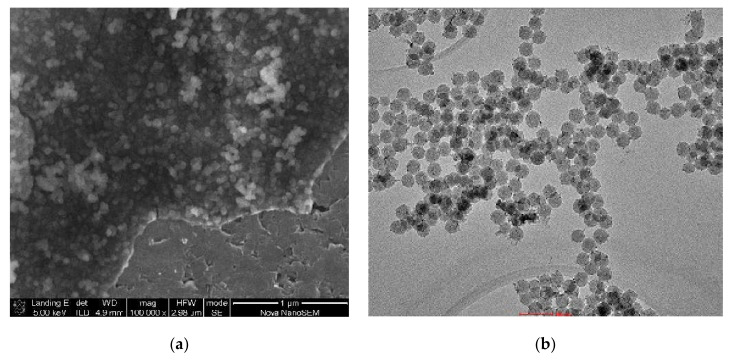
TEM image of PcZrGallate-SiO_2_ Nps obtained by the B-method (scale 100 nm) (**a**); SEM image of PcZrGallate-SiO_2_ Nps obtained by the B-method (scale 1 μm) (**b**).

**Figure 2 molecules-26-00260-f002:**
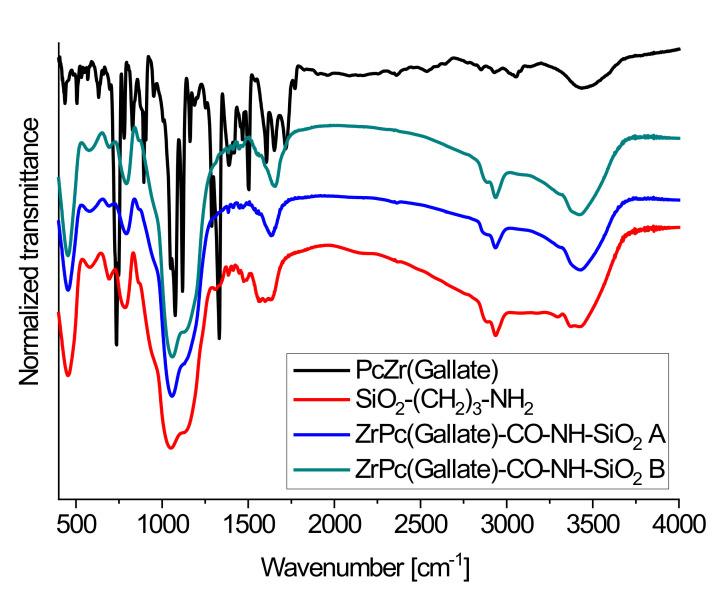
IR spectroscopy (MIR) spectra of PcZrGallate (black line), free alkylamino-modified SiO_2_ carrier (red line) and dye-modified carriers obtained by the **A** (blue line) and **B** (green line) methods.

**Figure 3 molecules-26-00260-f003:**
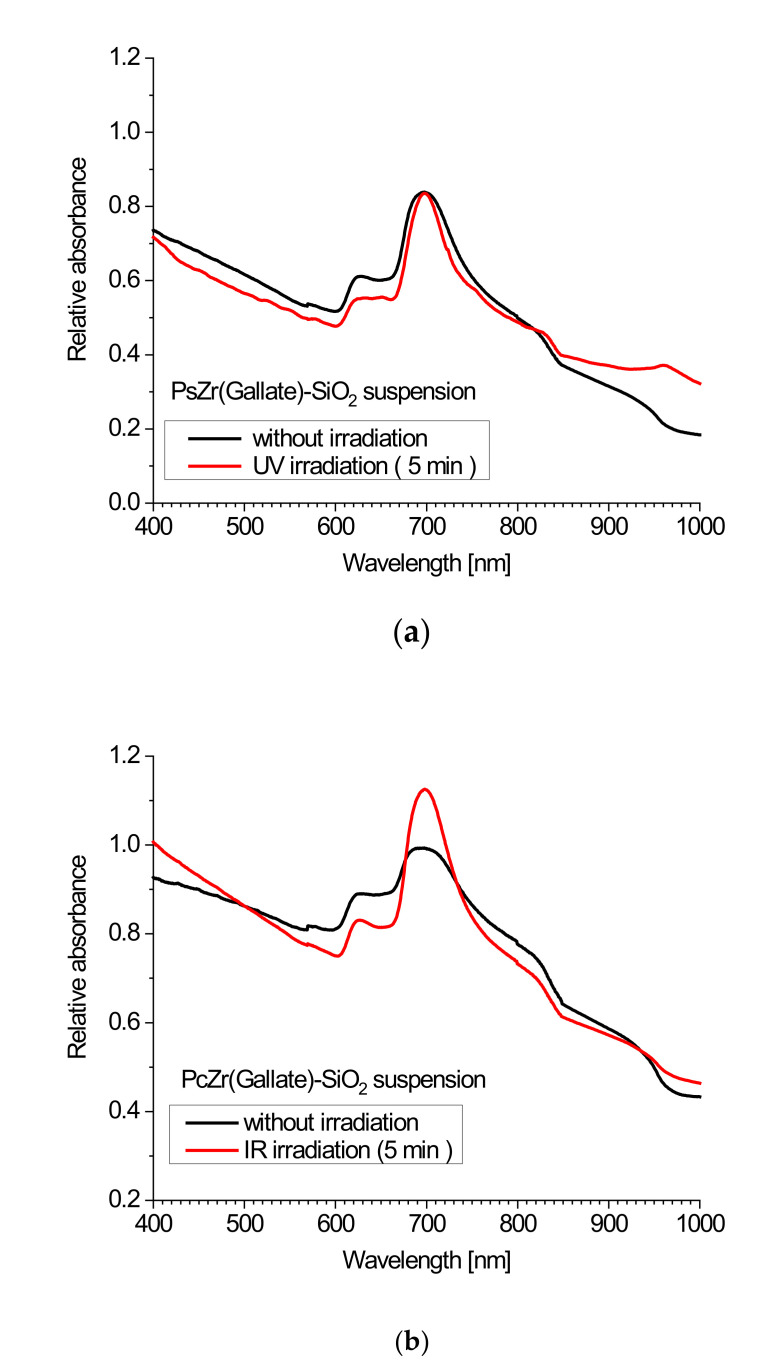
The absorption spectra of composites obtained by the A method before and after irradiation with a wide range 150 W UV lamp (**a**) and 150 W IR lamp (**b**).

**Figure 4 molecules-26-00260-f004:**
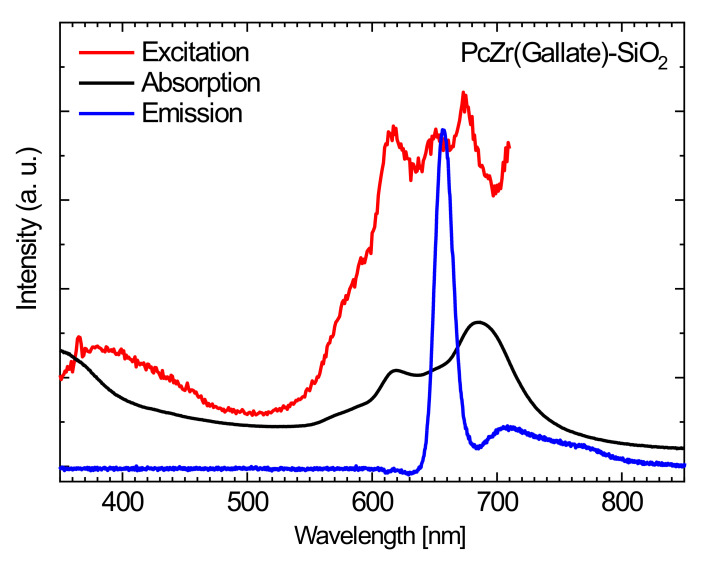
The absorption (black line), excitation (red line) and emission (blue line) spectra of the stable dispersion of conjugate PcZr(Gallate)-SiO_2_ in saline.

**Figure 5 molecules-26-00260-f005:**
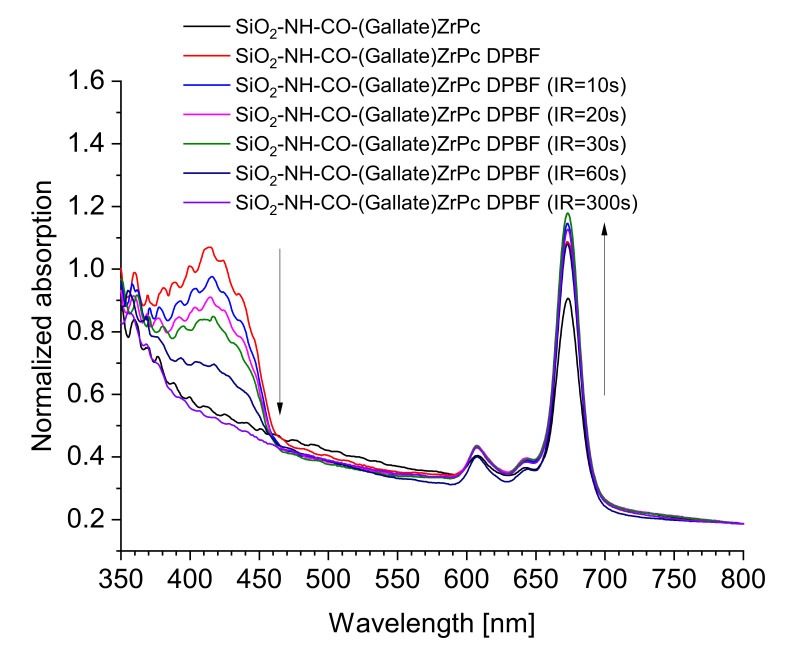
The ROS generation experiment for the PcZr(IV)gallate@SiO_2_ composite suspension in DMSO in the presence of 1,3-diphenylisobenzofuran (DPBF).

**Figure 6 molecules-26-00260-f006:**
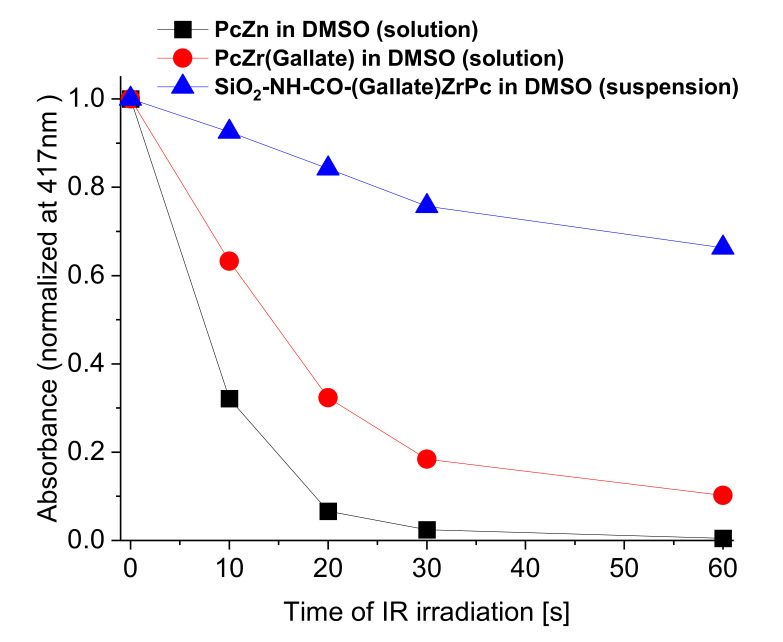
The dependence of the absorbance intensity (bleaching) of indicative pigment in the DPBF band maximum (λ_max_ = 417 nm) on the red light irradiation time.

**Figure 7 molecules-26-00260-f007:**
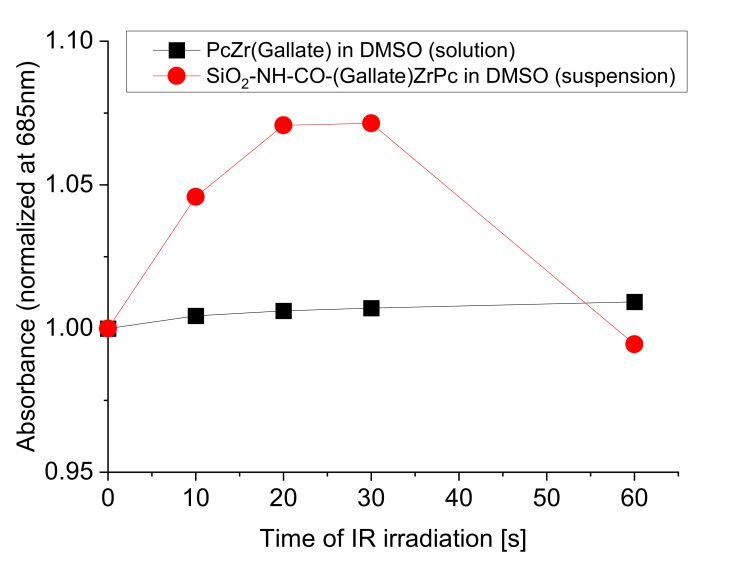
The dependence of the absorbance intensity in the Q band maximum of the PcZr (Gallate) complex at 685 nm on the exposure time with red light, for the complex solution and for the composite suspension PcZr(Gallate)@SiO_2_ in DMSO.

**Figure 8 molecules-26-00260-f008:**
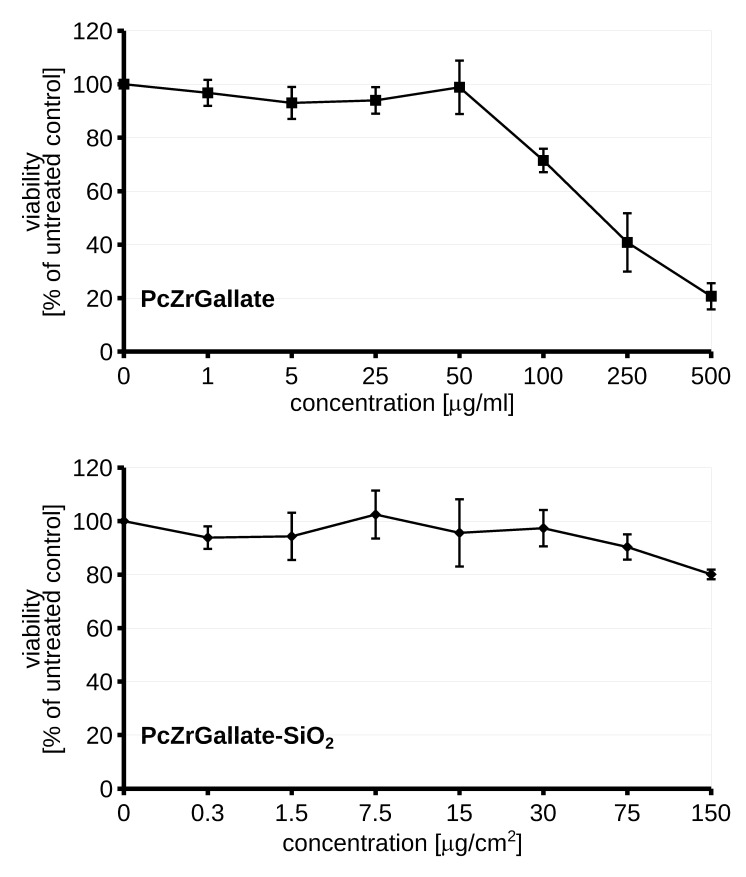
The dark cytotoxicity of the PcZrGallate and PcZrGallate SiO_2_nano RAW264.7 nanoformulations. The average viability along with SD were shown on the graph.

**Figure 9 molecules-26-00260-f009:**
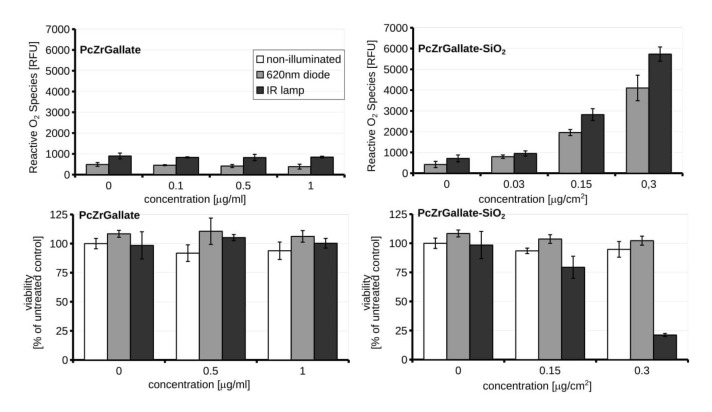
The induction of reactive O_2_ species upon illumination with RAW264.7 photosensitized with PcZrGallate or PcZrGallate SiO2 nanoformulations and accompanied by viability measurements illuminated cells. Averages ± SD were shown on the graph.

**Figure 10 molecules-26-00260-f010:**
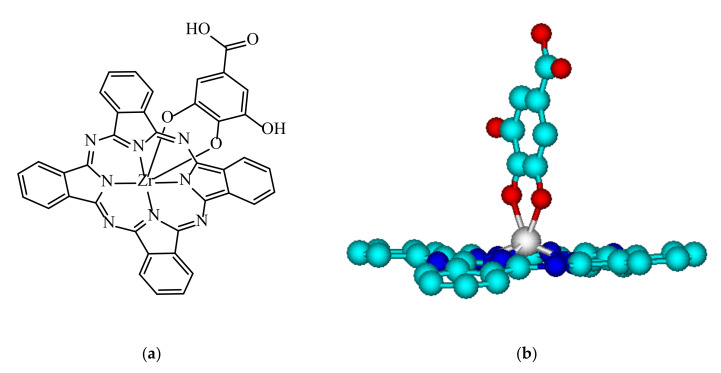
The structural formula of PcZr(Gallate) complex (**a**) and its 3D molecular structure model generated in the ACD Lab ChemSketch program (**b**).

**Figure 11 molecules-26-00260-f011:**
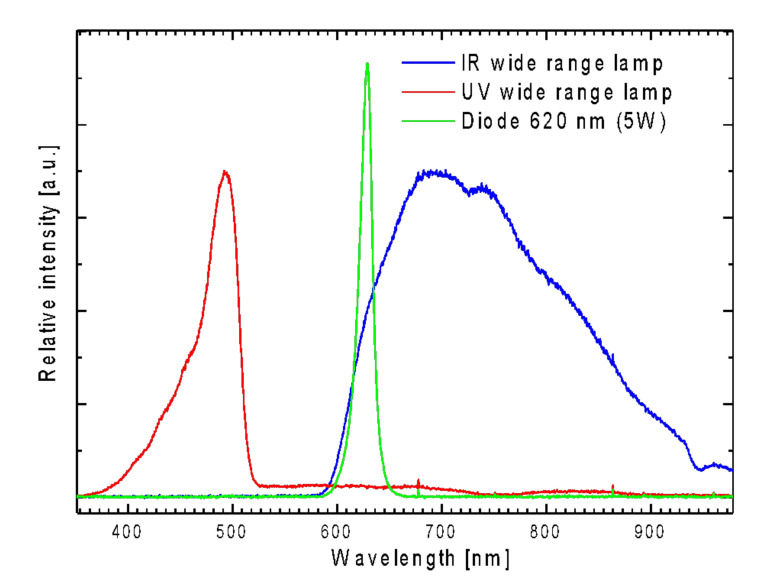
The emission spectra of irradiation sources: 150W UV MHN-TD Philips lamp (red line), 5w 620 nm THOR laser diode (green line), 150W Philips wide range red lamp (blue line).

## Data Availability

The data presented in this study are available on request from the corresponding author.

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
