# Peer review of "Gallato Zirconium (IV) Phtalocyanine Complex Conjugated with SiO2 Nanocarrier as a Photoactive Drug for Photodynamic Therapy of Atheromatic Plaque"

_molecules, 2021, doi:10.3390/molecules26020260_

Round 1
Reviewer 1 Report
This paper reports the synthesis of a new axially gallato substituted zirconium phthalocyanine complex and its evaluation as a PDT agent for treating athlerosclerosis. Overall, the application is interesting, but given the results reported, it is not clear how feasible it will be.
The syntheses presented in the manuscript are relatively straightforward and present minor incremental improvements over similar syntheses involving axial ligand substitutions/nanocarrier formulations with phthalocyanines.
The charaterization of the compounds is sufficient and well presented, though the discussion of each piece of data could be improved by condensing the writing and by proofreading what is written.
The PDT experiments must be improved. First, only once cell line is used in the experiments presented in the manuscript. I would expect at least one positive or negative control line (preferably both) to be used in this study. For example, the use of a non-phagocytic control line would help better understand selectivity issues.
In addition, cell studies should involve Pc alone, Pc-conjugate, SiO2 carrier alone, and SiO2 conjugate. This would help to ensure that any observed toxicity is due to activation of the Pc under the reported PDT conditions.
Finally, and perhaps most importantly, if the nanoparticles are not soluble and simply sink to the bottom of the cell culture wells, the results are inconclusive at best. Is the observed decrease in viability a proximity effect or is it due to internalization of the particles by the phagocyctic cells? This is a reasonable experiment to do, at least on a qualitative level.
Overall, there is still a lot of work to be done here with respect to the PDT studies and these studies should be done before publication.
A couple of other considerations:
- The introduction of the paper is very long and reads more like a review. This should be shortened to highlight relative aspects of previous works and to put the current work into perspective.
- There are a lot of issues with the writing including incorrect tenses, missing words, incorrect word usage, and words strung together without spaces. At times, this made the manuscript very difficult to read. These issues need to be addressed prior to publication.
Author Response
Reviewer 1.
This paper reports the synthesis of a new axially gallato substituted zirconium phthalocyanine complex and its evaluation as a PDT agent for treating athlerosclerosis. Overall, the application is interesting, but given the results reported, it is not clear how feasible it will be.
The syntheses presented in the manuscript are relatively straightforward and present minor incremental improvements over similar syntheses involving axial ligand substitutions/nanocarrier formulations with phthalocyanines.
The charaterization of the compounds is sufficient and well presented, though the discussion of each piece of data could be improved by condensing the writing and by proofreading what is written.
The PDT experiments must be improved. First, only once cell line is used in the experiments presented in the manuscript. I would expect at least one positive or negative control line (preferably both) to be used in this study. For example, the use of a non-phagocytic control line would help better understand selectivity issues.
In addition, cell studies should involve Pc alone, Pc-conjugate, SiO2 carrier alone, and SiO2 conjugate. This would help to ensure that any observed toxicity is due to activation of the Pc under the reported PDT conditions.
On a first glance I would agree with this comment. But at the end I must disagree. Regarding the SiO2 and the SiO2-conjugate control lines: the scheme proposed by the Reviewer would be correct if our consideration was dark cytotoxicity but not phototoxicity. The presented results show a low dark cytotoxicity of photoactive Zr-Pc modified SiO2 nanoparticles and indeed this is a control treatment for the phototoxicity test. We could add a control of SiO2 nanoparticles but it will show a lack of toxicity, as we and others (Bancos et al. (2012); Murugadoss et al. (2017)) have already shown. It would be like adding control to control treatment and so on. In result no new information would be obtained.
Finally, and perhaps most importantly, if the nanoparticles are not soluble and simply sink to the bottom of the cell culture wells, the results are inconclusive at best.
I’m really glad with this reviewer’s comment. Of course inorganic nanoparticles are not soluble and sink to the bottom of the cell culture (Böhmert et al.). And this applies to the vast majority of inorganic nanoparticles and similarly vastly underappreciated. Thus, a quantitative comparison of its action with soluble compounds is extremely difficult. But in our case we only observed the effect of nanoparticles and did not observe the effect of used soluble compound, so we can draw these conclusions.
Is the observed decrease in viability a proximity effect or is it due to internalization of the particles by the phagocyctic cells? This is a reasonable experiment to do, at least on a qualitative level.
This is a valid question. Regarding cytotoxicity of nanoparticles the internalization of particle is key event for the cytotoxic reaction (Sabella et al.; E. Wysokińska et al..; Edyta Wysokińska et.al.; Hsiao et al.). This is why the macrophage cell line was selected for the studies. Use of cell line with high phagocytosis capacity increases the sensitivity of experimental system on the level of phototoxicity, but also in means the dark cytotoxicity were low decrease of viability was observed even in high concentrations of nanoparticles.
We could think about the role of externally generated ROS in cytotoxicity as suggested by reviewer. The experimental setup would involve use of media depleted from ROS scavengers like glutmate or ascrobic acid, addition of ROS scavengers, usage of phagocytosis inhibitor (to compare sensitivity of same cells as the different cell lines have different toleration on ROS). But it would be a more mechanistic beyond presentation of activity of improved formulation of phototoxic material that is suitable for PDT.
Overall, there is still a lot of work to be done here with respect to the PDT studies and these studies should be done before publication.
A couple of other considerations:
- The introduction of the paper is very long and reads more like a review. This should be shortened to highlight relative aspects of previous works and to put the current work into perspective.
With great respect to the reviewer, the removal of any part of the introduction (whether in the chemical, biological or medical part) will contribute to the clarity of understanding of the general concept of the work.
- There are a lot of issues with the writing including incorrect tenses, missing words, incorrect word usage, and words strung together without spaces. At times, this made the manuscript very difficult to read. These issues need to be addressed prior to publication.
As far as possible, the content, technical and grammatical errors have been corrected.
- Bancos, S., et al. „Evaluation of Viability and Proliferation Profiles on Macrophages Treated with Silica Nanoparticles In Vitro via Plate-Based, Flow Cytometry, and Coulter Counter Assays”. ISRN Nanotechnology, 2012 (2012) 1–11. doi:10.5402/2012/454072.
- Böhmert, Linda, et al. „In vitro nanoparticle dosimetry for adherent growing cell monolayers covering bottom and lateral walls”. Particle and Fibre Toxicology, vol. 15 (1) (2018) 42. doi:10.1186/s12989-018-0278-9.
- Hsiao, I. Lun, et al. „Trojan-Horse Mechanism in the Cellular Uptake of Silver Nanoparticles Verified by Direct Intra- and Extracellular Silver Speciation Analysis”. Environmental Science & Technology, 49 (6) (2015) 3813–21. doi:10.1021/es504705p.
- Murugadoss S. et al. „Toxicology of Silica Nanoparticles: An Update”. Archives of Toxicology, 91 (9) (2017) 2967–3010. doi:10.1007/s00204-017-1993-y.
- Sabella S. et al. „A General Mechanism for Intracellular Toxicity of Metal-Containing Nanoparticles”. Nanoscale, 6 (12) (2014) 7052. doi:10.1039/c4nr01234h.
- Wysokińska et al., „Cytotoxic Interactions of Bare and Coated NaGdF4:Yb(3+):Er(3+) Nanoparticles with Macrophage and Fibroblast Cells”. Toxicology in Vitro: An International Journal Published in Association with BIBRA, 32 (2016) 16–25. doi:10.1016/j.tiv.2015.11.021.
- Wysokińska E. et al.,. „Toxicity Mechanism of Low Doses of NaGdF4:Yb3+,Er3+ Upconverting Nanoparticles in Activated Macrophage Cell Lines”. Biomolecules, 9 (1) (2019) 14. doi:10.3390/biom9010014.
Reviewer 2 Report
The paper describes alkylamino Zr-Pc modified SiO2 nanoparticle for PDT applications in artery diseases.
Overall, the paper the manuscript needs in depth grammar, and format revision. Many misspellings, and specially, in many sentences, words that are written together!! Did any of the authors give a view before submitting the manuscript?
Particularly:
- For a correct characterization of the composite material, the real content (numerical relative content) of the Gallate-ZrPc in the nanoparticles must be evaluated by optical, or XPS, or thermogravimetric means
- When describing the 1HNMR, coupling constants should be included
- The authors state that the characteristic vibrations of the amide appear in the IR spectra (FIGURE 4) of the modified nanoparticles, but they do not. Another Figure where the four IR spectra can be clearly distinguished (not overlapped) and whereby the bands of the amide bond in the functionalized nanoparticles are visible. This is the way to confirm the covalent functionalization. Actually, the three IR spectra of the silica nanoparticles, functionalized and not, look pretty much the same
-The authors intend to do a qualitative comparison of the Pc content in Figure 5. For that, normalized absorption of both materials in the same graph should be shown. But again, I insist that a qualitative analysis of the content should be done.
Author Response
Reviewer 2
The paper describes alkylamino Zr-Pc modified SiO2 nanoparticle for PDT applications in artery diseases.
Overall, the paper the manuscript needs in depth grammar, and format revision. Many misspellings, and specially, in many sentences, words that are written together!! Did any of the authors give a view before submitting the manuscript?
Particularly:
- For a correct characterization of the composite material, the real content (numerical relative content) of the Gallate-ZrPc in the nanoparticles must be evaluated by optical, or XPS, or thermogravimetric means
Unfortunately, in such a short period of time and in the currently difficult access to laboratories due to restrictions caused by the COVID-19 pandemic, we are not able to carry out these additional studies proposed by the Reviewer for this work. However, the phthalocyanine content of the conjugate obtained by the method A was determined by ICP-OES and the results were already included in the text.
- When describing the 1HNMR, coupling constants should be included
The required data are implemented in the section "Materials and methods".
- The authors state that the characteristic vibrations of the amide appear in the IR spectra (FIGURE 4) of the modified nanoparticles, but they do not. Another Figure where the four IR spectra can be clearly distinguished (not overlapped) and whereby the bands of the amide bond in the functionalized nanoparticles are visible. This is the way to confirm the covalent functionalization. Actually, the three IR spectra of the silica nanoparticles, functionalized and not, look pretty much the same
Silica has very intensive band in the region of 500-1500 cm-1. Figure 4 shows that phthalocyanine also has signals in this region. Since phthalocyanine is applied only to the SiO2 surface, its concentration is very low compared to SiO2 and is not evident in the presented spectrum. The presence of phthalocyanine on the surface is well illustrated by the EAS spectra, which we present in our work.
Unfortunately, due to the low concentration of the phthalocyanine complex in conjugate and the way it is distributed in conjugate (on the surface or in pores), the characteristic vibrations of amide groups in the range above 3000 cm-1 and below 800 cm-1 are not identifiable. On the other hand, based on data presented by Naz et al. [1] we know that two bands at around 1490 and 1560 cm-1, and a shoulder at 1630 cm-1 are correspond to APTES-modified silica. In addition, the IR spectrum of standard gallic acid also shows three bands in this region with maxima at 1536, 1615, 1709 cm-1 Zhang D. et al. [2] and ICD FTIR DATABASE [3]. It is also known that a metal coordinated phthalocyanine ring has a characteristic band at 1608-1610cm-1 (Verma et al. [4] and ACD FTIR data base [3]) and the amide bond is responsible for various characteristic vibrations depends on conformation (Yan Ji et al. [5]). Due to the low concentration of phthalocyanine complex in the conjugate, the characteristic amide-related bands in 3300-3500 cm-1, and in 400-900 cm-1 region are not detected. However we can clearly observe a wide intensive band in the range 1630-1680cm-1.
In our work, for the conjugate A we observe a wide band with a maximum at 1633cm-1 followed by a shoulder at 1657cm-1, while for the conjugate B the situation is the opposite: we see a wide, intensive signal at 1656cm-1 and then a wide shoulder at 1630cm-1. We agree that the band at ~1630 cm-1 can be associated with modified silica, but the band at 1656cm-1 most likely corresponds to the C=O stretching vibration in "Amide I" type of conformation of amide group by Yan Ji et al. [5] classification. An incorrect wavenumber value in the text of the article has been also corrected.
- Saba Naz et al. Synthesis of Highly Stable Cobalt Nanomaterial Using Gallic Acid and Its Application in Catalysis, Advances in Chemistry 2014 (2014) 686925, DOI: 10.1155/2014/686925
- Dezhi Zhang et al. Immobilization of cellulase on a silica gel substrate modified using a 3‑APTES self‑assembled monolayer, SpringerPlus (2016) 5:48 DOI 10.1186/s40064-016-1682-y
- A.L. Verma et al. Hydrogen peroxide vapor sensor using metal-phthalocyanine functionalized carbon nanotubes Thin Solid Films 519 (2011) 8144–8148 DOI:10.1016/j.tsf.2011.06.034
- ACD Spectrus Processor 2020 with internal FTIR-Raman spectra database (Advanced Chemistry Development, Ontario, Canada)
- Yan Ji et al. DFT-Calculated IR Spectrum Amide I, II, and III Band Contributions of N-Methylacetamide Fine Components ACS Omega. 5 (15) (2020) 8572–8578. DOI: 10.1021/acsomega.9b04421
-The authors intend to do a qualitative comparison of the Pc content in Figure 5. For that, normalized absorption of both materials in the same graph should be shown. But again, I insist that a qualitative analysis of the content should be done.
As a result of technical problems that occurred while sending the article to the journal (Figure 5 did not appear in the editorial version), it was changed to an incorrect description of the figure as well. The figure description was changed to the correct one in the current version. Due to the fact that the absorption measurements were carried out for different radiation sources, their normalization and placing on one drawing is considered inappropriate. Quantitative determination of phthalocyanin complex content in conjugate was carried out by us and the data was included in the text.
Reviewer 3 Report
The manuscript: ID molecules-1035085, by Yuriy Gerasymchuk et al, entitled "Gallato zirconium (IV) phtalocyanine complex conjugated with SiO2 nanocarrier as photoactive drug for photodynamic therapy of atheromatic plaque", refers to the characterization of the photochemical properties of composite material in saline environment sunder irradiation, with different light sources. Moreover, the reactive oxygen species (ROS) generation in DMSO suspension, under near IR irradiation, was also evaluated. the authors found that the PcZrGallate-SiO2
conjugate induces the cytotoxic effect on macrophages after irradiation with IR light, an effect that did not correspond to ROS production. SiO2, as a carrier, enables the photosensitizer to enter the macrophage, a type of cell that plays a key role in development of atheroma.
These properties have the perspective to render the conjugate, they developed, useful in the photodynamic therapy of coronary artery
disease.
From this point of view, the present article is very interesting. However, the revised version, should be proof-read by an English speaking person.
Author Response
Reviewer 3
The manuscript: ID molecules-1035085, by Yuriy Gerasymchuk et al, entitled "Gallato zirconium (IV) phtalocyanine complex conjugated with SiO2 nanocarrier as photoactive drug for photodynamic therapy of atheromatic plaque", refers to the characterization of the photochemical properties of composite material in saline environment sunder irradiation, with different light sources. Moreover, the reactive oxygen species (ROS) generation in DMSO suspension, under near IR irradiation, was also evaluated. the authors found that the PcZrGallate-SiO2
conjugate induces the cytotoxic effect on macrophages after irradiation with IR light, an effect that did not correspond to ROS production. SiO2, as a carrier, enables the photosensitizer to enter the macrophage, a type of cell that plays a key role in development of atheroma.
These properties have the perspective to render the conjugate, they developed, useful in the photodynamic therapy of coronary artery disease.
From this point of view, the present article is very interesting. However, the revised version, should be proof-read by an English speaking person.
We are grateful to the reviewer for the positive assessment of our work. As far as possible, the content, technical and grammatical errors have been corrected.
Round 2
Reviewer 2 Report
The issues have been correctly addreses